# Effect of Cd on Mechanical Properties of Al-Si-Cu-Mg Alloys under Different Multi-Stage Solution Heat Treatment

**DOI:** 10.3390/ma15155101

**Published:** 2022-07-22

**Authors:** Hongkui Mao, Xiaoyu Bai, Feng Song, Yuewen Song, Zhe Jia, Hong Xu, Yu Wang

**Affiliations:** School of Materials Science and Engineering, North University of China, Taiyuan 030051, China; maohk@126.com (H.M.); baixiaoyu2021@126.com (X.B.); ufghyj@126.com (F.S.); songy0612@163.com (Y.S.); zhejia@zju.edu.cn (Z.J.)

**Keywords:** Al-Si-Cu-Mg cast alloys, Cd, multi-stage heat treatment, precipitation

## Abstract

Multi-stage heat treatment is an important method to improve the mechanical properties of Al-Si-Cu-Mg aluminum alloys. In this paper, the multi-stage heat treatment was carried out for the Cd-free and Cd-containing alloys. The experimental results show that the addition of Cd promoted the precipitation of Q″ and θ″, which led to the formation of a large number of fine, dispersed precipitates and a higher yield strength (YS) and ultimate tensile strength (UTS) for the Cd-containing alloys. The addition of Cd also altered the optimal heat treatment parameters. For the Cd-free alloys, the Cu-rich phase fully dissolved after three-stage heat treatment, and the YS and UTS of the three-stage heat-treated alloys were higher than their two-stage heat-treated counterparts. For the Cd-containing alloys, the three-stage heat treatment led to the precipitation of Cd-rich low melting point phases, caused defects, and reduced the mechanical properties of the alloy. The size and volume fractions of the precipitates were significantly less than those of the alloys after two-stage heat treatment and the strength of the alloys decreased. Therefore, the solution time should be strictly controlled for Cd-containing Al-Si-Cu-Mg alloys.

## 1. Introduction

Al-Si-Cu-Mg alloys are typical heat-treatable aluminum alloys whose strength could be improved by heat treatment [1,2,3,4], and have been widely used in the aerospace and transportation industries [5,6]. The mechanical properties of the alloys mainly depend on the alloys’ composition and processing parameters. To meet the increasingly demanding requirements, the mechanical properties of cast alloys’ parts were further improved by microalloying and fine-tuning the parameters of the heat treatment process.

In recent decades, it has been found that adding Cd [7,8,9,10,11,12,13], In [7,8] and Sn [7,8,14,15,16] to Al-Cu alloys and adding Cu [17,18,19], Ge [20], and Ag [19] to Al-Mg-Si alloys could significantly promote the aging kinetics and improve the strength of the alloys. Microalloying improves the strength of the alloys by producing refined and homogeneous distributed strengthening phases with a higher density. Since the formation of precipitates was controlled by vacancy diffusion, the high-vacancy binding energy of trace elements had a decisive influence on the precipitated structures that were formed during the aging process. Taking the θ phase as an example, the main precipitation sequence of Al-Si-Cu-Mg alloys was as follows: α_sss_ → Guinier-Preston (GP) zone → θ″ phase (nominal stoichiometry Al_3_Cu with a tetragonal crystal structure: a = b = 0.398 nm, c = 0.766 nm) → θ′ phase (nominal stoichiometry Al_2_Cu with a tetragonal crystal structure: a = b = 0.404 nm, c = 0.580 nm) → θ phase [21,22,23]. The θ″ and θ′ phases were coherent with the aluminum matrix [23,24]. Hardy et al. [7] and Runxia et al. [9] studied the effect of Cd on the aging process of Al-Si-Cu-Mg alloy cast aluminum alloy and found that adding Cd to the Al-Si-Cu-Mg alloy could accelerate the aging process of the alloy, and with the increase of Cd addition. The strength of the Al-Si-Cu-Mg alloy showed a trend of first increase and then decrease, and the best effect was achieved when the Cd content was 0.3% [9,25]. But no one has studied the effect of Cd addition on the solid solution process of Al-Si-Cu-Mg alloy. Therefore, it was necessary to study the effect of Cd on Al-Si-Cu-Mg alloys at different solution temperatures and times.

Many studies had found that parameters such as solution time, aging time, and aging temperature had a great influence on the microstructure and properties of aluminum alloys [26,27,28]. Solid solution treatment was typically divided into a single-stage solid solution, a double-stage solid solution, and a multi-stage solid solution. The temperature of the single-stage solid solution was below the eutectic temperature, the phase with a higher melting point could not be completely dissolved, and the multi-stage solution treatment could lead to more complete dissolution of second phases and increase the strength of the aged alloys [29,30,31]. Sokolowski et al. [32] studied the possibility of a two-stage solution treatment of Al-Si-Cu-Mg alloys. First, the solution treatment was carried out for 8 h at a low temperature of 495 °C to dissolve the Cu-rich phase, and then the temperature was increased to 520 °C with a holding time of 2 h to homogenize the alloys’ elements and improve the strength and ductility. Han et al. [33] performed a single-step and multi-step solution heat treatment on Al-Si-Cu-Mg alloys and found that the use of multi-step solution treatment could lead to the fuller dissolution of the Al_2_Cu phase and better spheroidization of the Si particles, especially the modified alloys that were treated by a multi-stage solution treatment (the ultimate tensile strength value of Al-Si-Cu-Mg alloys reaches 434 MPa after adding Sr). So far, although many works had been done on multi-stage solution treatments, there was no report on the microstructure and mechanical properties of Cd containing Al-Si-Cu-Mg alloys that were subjected to different solution heat treatment procedures.

This study took the Al-6.5Si-1.5Cu-0.25Mg alloy as the research object. The parameters of the two-stage solution treatment were selected according to Aguilera Luna et al. [4], being those for a three-stage solution treatment on the heat treatment process for production. The effect of different solution heat treatments on the microstructure and mechanical properties of Al-6.5Si-1.5Cu-0.25Mg and Al-6.5Si-1.5Cu-0.25Mg-0.3Cd aluminum alloys was analyzed, with the optimal heat treatment parameters for the alloys identified.

## 2. Experimental Procedure

The Al-6.5Si-1.5Cu-0.25Mg, Al-10Cd, and Al-10Sr master alloys were used to prepare the alloys that were used in this study. The alloys were melted in a resistance furnace at 740 °C, Al-10Cd and Al-10Sr master alloys were added, and the melt was stirred mechanically after holding. C_2_Cl_6_ was used for degassing. After the temperature of the melt had been lowered to 720 °C, the alloy was poured into a steel mold that was preheated to 250 °C. The composition of the alloys was measured by SpectroMAXx LMF15 (SPECTRO Analytical Instruments GmbH, Kleve, Germany) and the results are shown in Table 1.

Vickers hardness was measured with a load of 25 g and a resting time of 20 s to track the age-hardening behavior of the samples. The solution treatment was carried out in a box furnace with a temperature control accuracy of ±1 °C. Immediately after the solution treatment, the aging treatment was carried out in a drying oven, and at least 4 samples were used for each heat treatment. The heat treatment parameters that were used in this experiment are shown in Table 2. The specimens for microstructural characterization were prepared using standard metallographic sample preparation techniques. The samples were corroded with Keller’s solution and characterized by Zeiss Axio Scope A1 (Carl Zeiss AG, Oberkochen, Germany) an optical microscope (OM) and a JEOL JSM 7200F (JEOL, Tokyo, Japan) scanning electron microscope (SEM). The alloys’ precipitates were observed with FEI Tecnai G2 F20 (FEI Company, Hillsboro, OR, USA) transmission electron microscope (TEM) and analyzed with an energy dispersive spectrometer (EDS). The tensile test was carried out on an Instron3382 electronic tensile machine (Instron Corporation, Boston, MA, USA). The tensile sample was machined according to GB/T228.1-2010. The initial strain rate was 2.38 × 10^−4^ s^−1^. The reported value was the average of at least five measurements for each test condition.

## 3. Results

### 3.1. Microstructure

Figure 1 shows the microstructures of the as-cast Cd-free and the Cd-containing alloys. It can be seen that there was almost no change in the microstructure of the Cd-containing alloy and the Cd-free alloy. Figure 2 shows the microstructure of the Cd-free and Cd-containing alloys after solution heat treatment. For the two-stage heat treatment, the eutectic Si phase of the Cd-free alloys was spheroidized, and the Al_2_Cu phase was dissolved into the aluminum matrix as shown in Figure 2a,b, which shows that the eutectic Si in the Cd-containing alloys was more rounded, dispersed, and numerous. Several tiny holes were observed in the matrix and the phenomenon of incipient melting appeared [34]. Runxia et al. [9] found that in addition to the solid solution of Cd in the α-Al matrix, a small amount of Cd would be slightly enriched at the edge of the Cu-rich phase, so these tiny pores might be related to the addition of Cd; with the increase of temperature and time, the Cu-rich phase was enriched with a low-melting Cd phase, the phenomenon of incipient melting occurred and produced several tiny pores in the alloy.

For the three-stage heat treatment, the size and distribution of eutectic Si in the Cd-free alloys were similar to those in Figure 2a, but there were few coarser Si particles in the alloys, and the eutectic Si was finer and rounder, as shown in Figure 2c. This indicates that the tertiary solution treatment was more beneficial in obtaining a uniform solid solution with maximum supersaturation. It can be seen from Figure 2d that the Si phase of the Cd-containing alloys was finer, and while the size of the micropores in the alloys was obviously reduced, the number increased. Compared with Figure 2b, some micropores appeared around the eutectic Si. It can be seen that when the three-stage solution treatment was adopted, the low-temperature solution treatment stage (490 °C-4 h) was beneficial to the decomposition of the Cu-rich phase, thus greatly reducing the size of the pores.

### 3.2. Aging Hardness

Figure 3 shows the age-hardening curves for Cd-free and Cd-containing alloys. The Vickers hardness curve presented a bimodal phenomenon with the prolonging of the aging time. The curves of the Cd-free and Cd-containing alloys had a similar trend. The hardness reached the first peak around the 12-h mark, hit a valley at 16 h, and reached the second peak at the 24-h point with hardness the same as that which was found at 12 h. The hardness of the Cd-containing alloy was higher than that of the Cd-free alloy with the same aging time, indicating that the addition of Cd improved the hardness of the alloy.

Figure 3 shows the hardness evolution of the alloy during aging at 175 °C. There were two aging peaks in the aging hardening process of Al-Si-Cu-Mg alloy, the GP region and the metastable phase could effectively strengthen the alloy and led to the aging peak. In the early stage of alloy aging, the fine and abundant GP region was distributed in the supersaturated solid solution (matrix) of the alloy, which played a strengthening role. In the middle stage of aging, the formed metastable phase that maintains a semi-coherent phase with the matrix effectively resisted the movement of dislocations and had a certain strengthening effect [9,10]. The GP region was known to dissolve significantly before the formation of the metastable phase, and θ phase had been observed to nucleate on dislocations [35]. Therefore, in the stage of transition from the GP region to the metastable phase, the number of GP regions was significantly reduced and dissolved, while the metastable precipitates did not grow up and were too small in size to effectively resist the movement of dislocations, resulting in a low age-hardening effect between the two aging peaks of the Al-Si-Cu-Mg alloy. According to the aging hardening curve, there is a bimodal aging result, and combined with the research results of Runxia et al. [25], it is concluded that the optimal aging process in this experiment is 175 °C × 12 h.

β (Mg_2_Si), θ phase (Al_2_Cu), and Q phase (Al_5_Cu_2_Mg_8_Si_6_) might appear in Al-Si-Cu-Mg alloys. The research of Qiao Xiao et al. [36] showed that the Cu element would inhibit the precipitation of the β phase during the aging process, when Cu/Mg ≈ 2.1, the β′ phase in the alloys could be completely transformed into a ternary composite phase of (α + Si + Q); when Cu/Mg > 2.1, the θ phase would also appear. The composition of the alloys in this paper was Al-7Si-1.5Cu-0.25Mg, Cu/Mg ≈ 6, and there was no β phase precipitation in the alloys. The precipitation strengthening phases of the alloys were θ and Q phases, and the θ phase was a lath-like precipitate, while the Q phase was a point block precipitate [37,38,39].

Figure 4 shows the microstructure of the Cd-free alloy after 2 h, 12 h, 24 h, and 48 h of aging. It could be seen from the figure that with the extension of time, the precipitates gradually nucleated and grew up, and the Q phase was a coarse precipitate, while the θ phase was a fine precipitation phase, the number of which was much larger than that of the Q phase. We could also see that in the process of precipitation, the growth rate of the Q phase was higher than that of the θ phase, which might be related to the diffusion rate of Si and Mg elements being higher than that of Cu atoms [1].

Figure 5 shows the morphology of the precipitated phases of Cd-containing and Cd-free alloys after two-stage solution treatment and an aging time of 12 h. It can be seen from Figure 5a that the number of precipitates in the Cd-containing alloys was significantly higher than that in the Cd-free alloys (Figure 5b), and the size of the precipitate was smaller. The addition of Cd atoms was beneficial in promoting the precipitation of Cu-containing phases such as θ and Q, resulting in a large number of precipitated phases in the alloys.

### 3.3. Tensile Properties and Fracture Analysis

Figure 6 shows the stress–strain curves of Cd-free/containing alloys after different heat treatments. Table 3 shows the mechanical properties of Cd-free and Cd-containing alloys after different heat treatments. The UTS and YS of the Cd-free alloys that were treated by the three-stage solution treatment were higher than those of the two-stage solution treatment; the UTS and YS of the Cd-containing alloys that were treated by the two-stage solution treatment were higher than those of the three-stage solution treatment. The UTS and YS of the Cd-containing alloys were higher than those of the Cd-free alloys, while the elongation was lower than that of the Cd-free alloys irrespective of the heat treatment procedures.

After the three-stage treatment, the Cd-free alloys had finer and more rounded eutectic Si phases than two-stage solution-treated alloys and during the three-stage solution treatment, the alloy was first held at 490 °C for 4 h, which helped to dissolve a part of Al_2_Cu in the matrix before raising the temperature to 520 °C, which accelerated the decomposition and diffusion of the remaining Cu-rich phase and further enhance the solute Cu in α-Al, so that more θ″ phase could be precipitated during aging, and the strengthening effect of the alloys could be improved [4]. It can be seen from Figure 5 that the size and volume fraction of the θ″ phase in the Cd-containing alloys after the two-stage solution treatment were significantly higher than those in the three-stage-Cd-containing alloys, which made the strengthening effect of the θ″ phase in the two-stage-Cd-containing alloys higher. In the end, the best heat treatment process was a two-stage heat treatment for the Cd-containing alloy.

Figure 7 shows the fracture morphologies of the Cd-free and Cd-containing alloys after the two-stage solution heat treatment and the 12-h aging treatment. Ductile fractures happened in all the alloys according to Figure 7; a large number of small and shallow dimples could be observed in Figure 7a, which was reflected in the fracture elongation of the alloy, reaching 11. Figure 7b shows that the local dimples were larger, the dimples were deepened, and the elongation rate was reduced to a certain extent. Figure 7c shows that part of the fracture mode was intergranular-fracture, and there were also a large number of dimples, and a white residual phase appeared at the bottom of the dimples. However, Figure 7d shows that a small amount of cleavage platform appears, the tear ridges were obvious, and the dimples became deeper, indicating that the matrix was effectively strengthened. There were more small white particles in Figure 7d than in Figure 7c. After EDS analysis, they were mainly Cd-rich phases, indicating that Cd-containing alloys would precipitate more Cd-rich phases with the prolongation of solid solution time. According to the research of Runxia et al. [9], that the Cd-rich phase was a low-melting phase. As the number of precipitates of the Cd-rich phase increased, it would reduce the grain boundary strength, which was not conducive to the strengthening of the alloy and reduced the mechanical properties, which was consistent with the results of tensile strength.

## 4. Discussion

### 4.1. Equilibrium Phase Diagram Analysis

To reveal the evolution of the phases in Al-6.5Si-1.5Cu-0.25Mg and Al-6.5Si-1.5Cu-0.25Mg-0.3Cd alloys with temperature, phase diagram calculations were performed using Thermo-Calc software that was based on the thermodynamic database TCAL4, and the results are shown in Figure 8. It showed that the solubility of Cd in the Al matrix reached a maximum of 0.27 wt.% at 540 °C and decreased gradually with decreasing temperature, and when the temperature was 200 °C, the solid solubility of Cd in Al was close to 0. Therefore, Cd atoms might be precipitated during heat treatment.

### 4.2. Low-Melting Cd Phase

Figure 9 shows the XRD patterns of the Cd-free and Cd-containing alloys before and after heat treatment. It can be seen that their peaks are not much different, and the diffraction peak of the intermetallic phase could be reasonably represented by Al, Si, and Al_5_Cu_2_Mg_8_Si_6_.

To analyze the precipitation of Cd during the solution treatment, the distribution of Cd elements in the alloys was analyzed by EDS. Figure 10 shows the SEM image of the Cd-containing alloys after two/three-stage solution treatment EDS results. It can be seen from the figure that the distribution of Cu elements was relatively uniform, and there was no obvious segregation, which is to say, the Cu-rich phase dissolved more completely; due to the low content of Cd, the EDS spectrum results showed that there was no obvious segregation of Cd elements. To further confirm the distribution of Cd elements, EDS point analysis was performed on the black area near Si, and it was found that the incipient melting phase was rich in Cd, which might be due to the high interfacial energy between Si and α-Al, which caused the Cd-rich low-melting phase to form easily near the Si phase.

Figure 11a shows the TEM image of the Cd-containing alloys after the two-stage solution treatment. The Si phase was surrounded by the Al phase. Small black particles appeared around the Si phase. According to the EDS results, the Cd element was only 0.8%. Figure 11b shows the TEM image of the Cd-containing alloys after the three-stage solution treatment. Small black particles appeared around the Si phase, and this phase was rich in Cd, with an atomic percentage of 11.91%, which confirmed that during the solution treatment, the Cd atoms would precipitate preferentially at the interface of the Si phase and Al phase. This was due to the high interface energy between the Si phase and the Al matrix, which provided nucleation sites for the Cd-rich phase. The enrichment of Cd atoms would reduce the Cd content in the α-Al phase, which would adversely affect the precipitation of the strengthening phases.

The impurity diffusion coefficient in solids is usually described by the Arrhenius equation: $D=D_{0}\exp\left(-\frac{Q}{\mathrm{RT}}\right)$ (cm^2^/s), where D_0_ denotes as the pre-exponential factor, Q the activation energy, T the absolute temperature, and R the gas constant [40,41,42]. D_(Cu/Al)_ = 0.06~20 cm^2^/s, Q_(Cu/Al)_ = 133.9~136.1 kJ/mole, D_(Cd/Al)_ = 1.04 cm^2^/s, Q_(Cu/Al)_ = 124.3 kJ/mole [43]. 

A longer solid solution time was more conducive to the homogenization of the Cu element and strengthens the alloys. However, if the solution time becomes longer, a Cd-rich low-melting phase will be formed, reducing the number of θ″ phases in the alloy and reducing the mechanical properties of the alloy. Therefore, it was very important to balance the solution treatment times. In the multi-alloys system, when the concentration ratio of Cu element and Cd element was different, it was difficult to keep the solid solution time in a relatively balanced state, which requires further research and is not discussed here.

To further verify the effect of the enrichment of the Cd atoms on the precipitation of the strengthening phases, the microstructure of the Cd-containing alloys after the two/three-stage solution heat treatment and aging were analyzed. Figure 12a shows the bright-field image of the Cd-containing alloys after the two-stage heat treatment. Figure 12b shows the bright-field image of the Cd-containing alloys after the three-stage heat treatment. There was also a certain number of precipitates that were evenly distributed in the matrix, but the number was slightly smaller. Figure 12c shows the high-resolution TEM of precipitates in the matrix of the Cd-containing alloy sample, and the results showed that the massive precipitates were evenly distributed in the matrix and the number was large, which could confirm that the precipitates were fine θ″ phase precipitates [43]. Comparing the TEM images of the Cd-containing alloys in the two-stage and three-stage heat treatments, it could be found that after the three-stage solution heat treatment of the Cd-containing alloys, the precipitates in the samples were significantly smaller in size and volume fraction, resulting in a lower strengthening effect than the two-stage solution treatment.

On the one hand, the formation of Cd-rich low-melting phase can lead to local defects. On the other hand, the precipitation of the Cd element would lead to delayed precipitation and the reduction of the number of θ″ phases near the Si phase, thereby reducing the aging precipitation strengthening ability, indicating that the Cd-containing alloys had a higher yield strength and tensile strength after the two-stage solution treatment.

## 5. Conclusions

Based on the present results, the following conclusions can be drawn:YS and UTS of Cd-containing alloys are stronger than Cd-free alloys, but the elongation is lower than the Cd-free alloys.The addition of Cd promotes the precipitation of the θ″ phase in the Al-Si-Cu-Mg alloy and improves the mechanical properties of the alloy.For the Cd-free alloys, a long solution time is conducive to the full diffusion and homogenization of copper elements, thereby improving the precipitation-strengthening ability of the alloy. For the Cd-containing alloys, the Cd element is enriched in the Si-Al phase boundary, which reduces the precipitation-strengthening ability of Cd, resulting in the alloys’ mechanical properties after a three-stage heat treatment being lower than that of a two-stage heat treatment.

## Figures and Tables

**Figure 1 materials-15-05101-f001:**
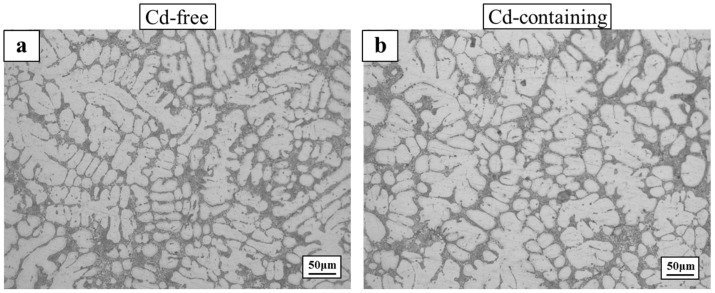
As-cast microstructure and metallographic images of the alloy: (**a**) Cd-free and (**b**) Cd-containing.

**Figure 2 materials-15-05101-f002:**
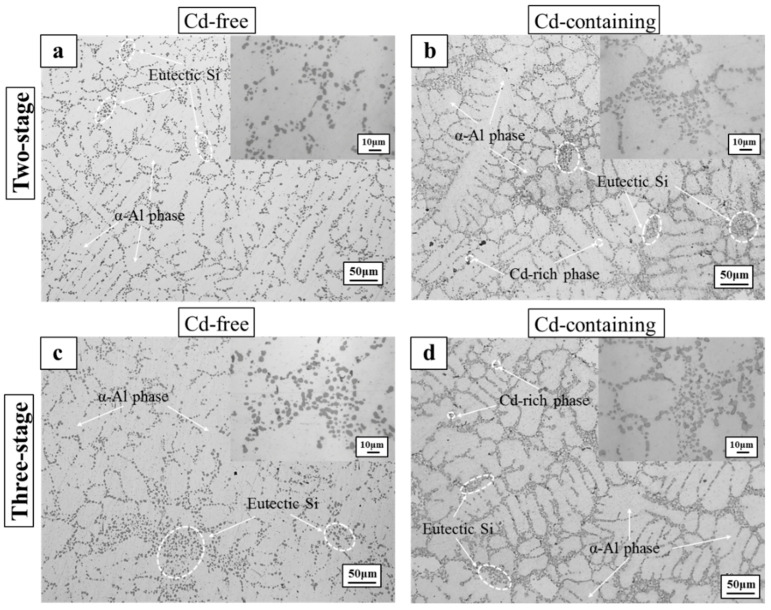
Microstructure and metallographic diagram of the alloys after solution treatment: (**a**) two-stage-Cd-free, (**b**) two-stage-Cd-containing, (**c**) three-stage-Cd-free, and (**d**) three-stage-Cd-containing.

**Figure 3 materials-15-05101-f003:**
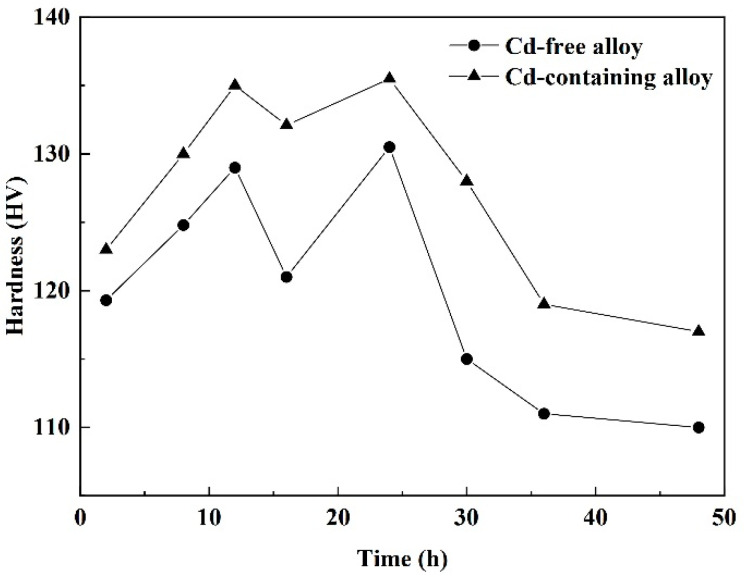
Hardness evolution of the alloys during aging at 175 °C.

**Figure 4 materials-15-05101-f004:**
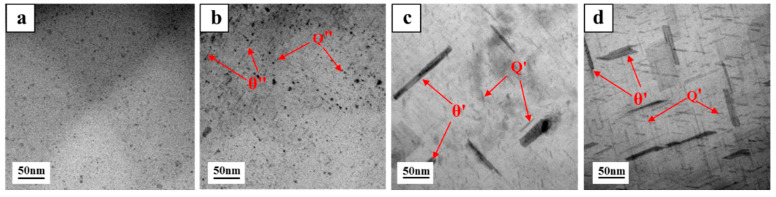
Precipitate in the Cd-free alloy after aging treatment: (**a**) 2 h; (**b**) 12 h; (**c**) 24 h; (**d**) 48 h.

**Figure 5 materials-15-05101-f005:**
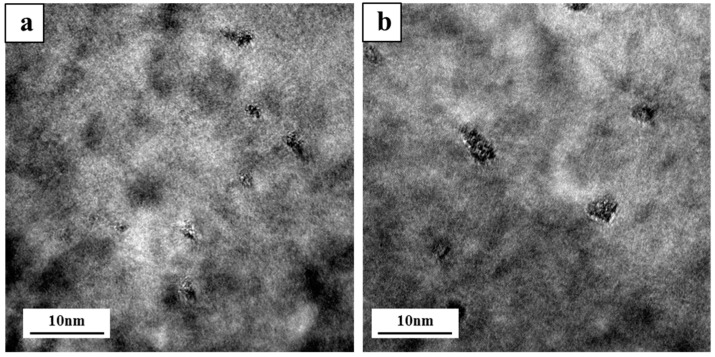
Precipitates after two-stage solution treatment and aging treatment (12 h), (**a**) Cd-containing alloy and (**b**) Cd-free alloy.

**Figure 6 materials-15-05101-f006:**
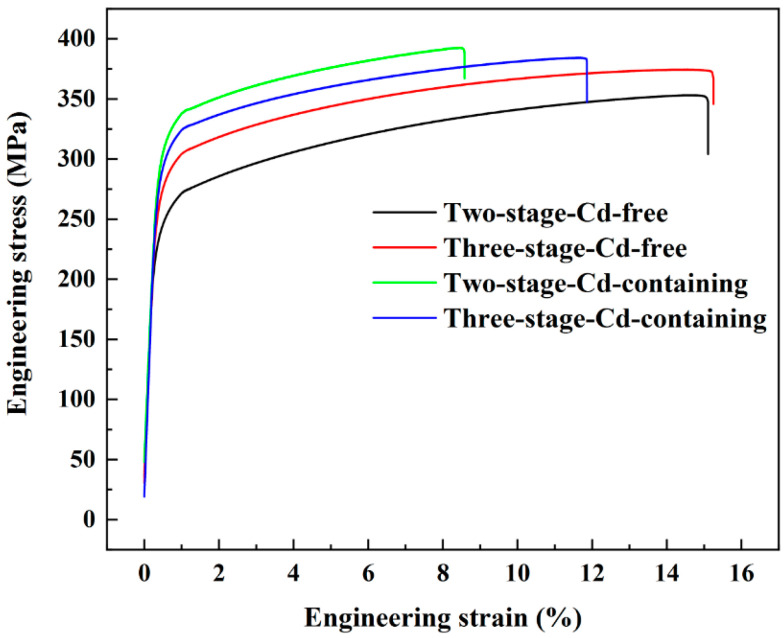
Stress-strain curves of the Cd-free and Cd-containing alloys after two-stage and three-stage heat treatment.

**Figure 7 materials-15-05101-f007:**
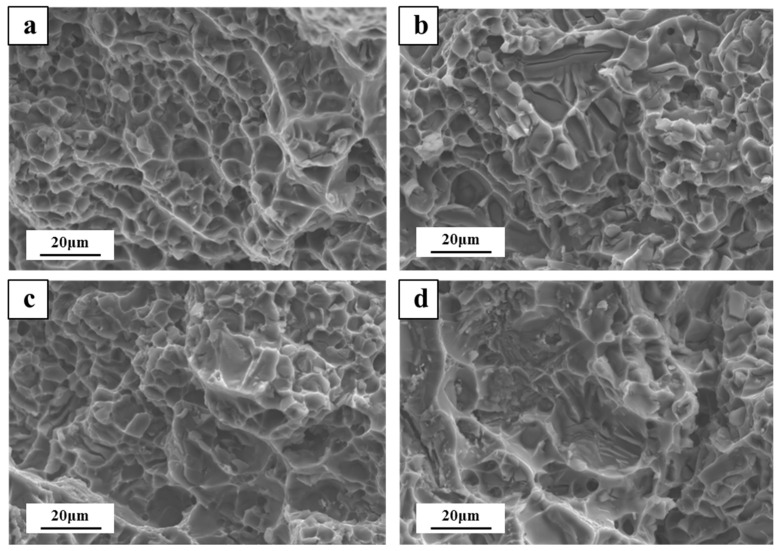
Fractography of the alloys after solution treatment: (**a**) two-stage-Cd-free, (**b**) two-stage-Cd-containing, (**c**) three-stage-Cd-free, (**d**) three-stage-Cd-containing, and (**e**) EDS analysis of small particles, (**f**) EDS analysis results of small particles.

**Figure 8 materials-15-05101-f008:**
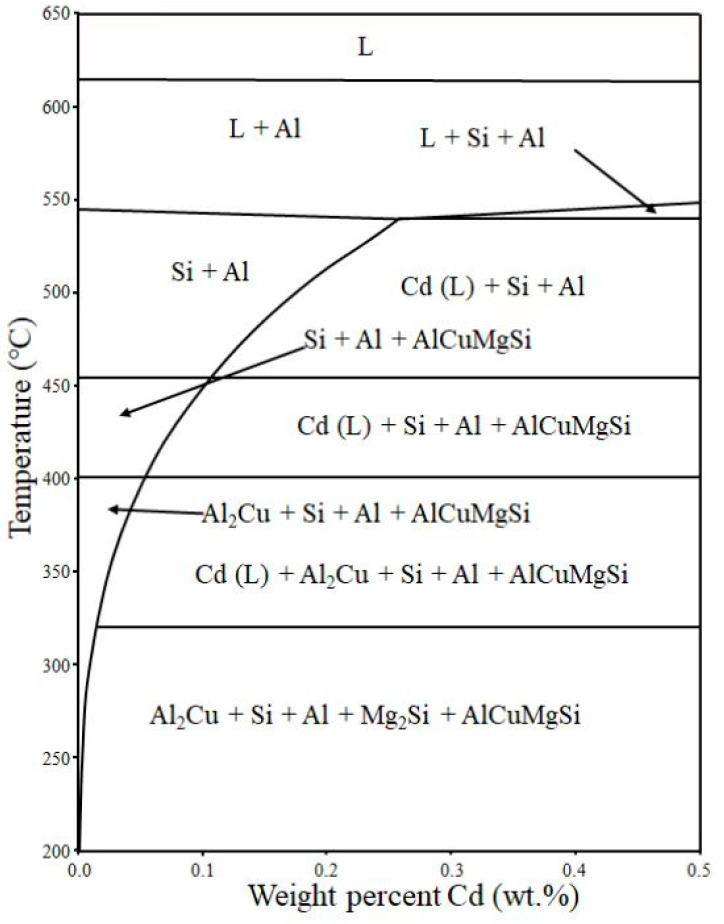
Calculated phase diagram showing the solubility of Cd in the Al-6.5Si-1.5Cu-0.25Mg matrix of the Al alloy as a function of temperature. In the phase diagram, L is liquid.

**Figure 9 materials-15-05101-f009:**
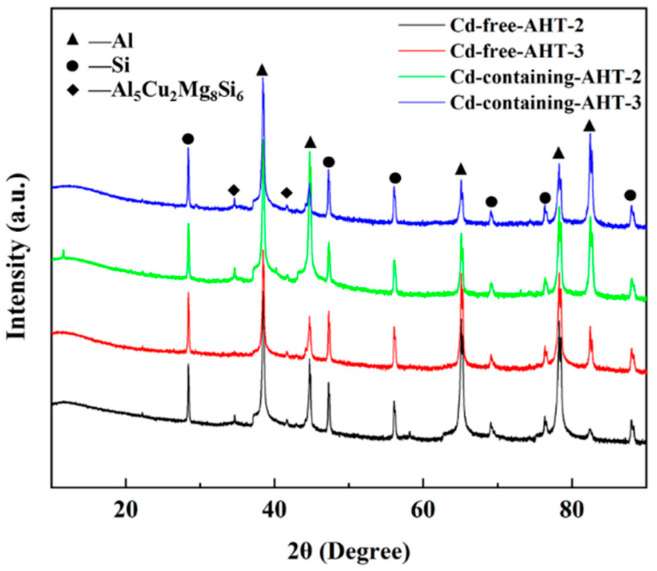
XRD patterns of Cd-free and Cd-containing after different heat treatment.

**Figure 10 materials-15-05101-f010:**
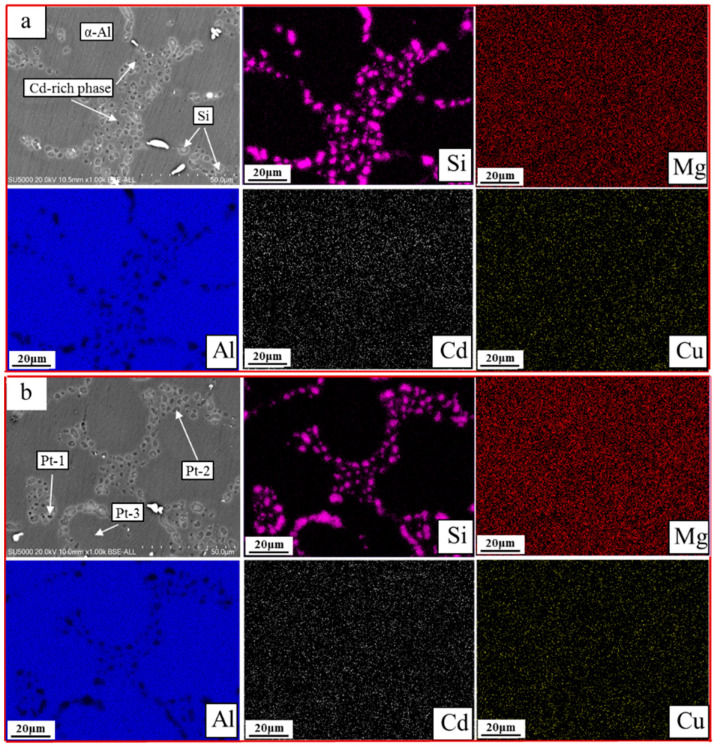
Characteristic SEM image, compositional mapping, and compositional point: (**a**) two-stage cadmium containing alloys and (**b**) three-stage cadmium containing alloys.

**Figure 11 materials-15-05101-f011:**
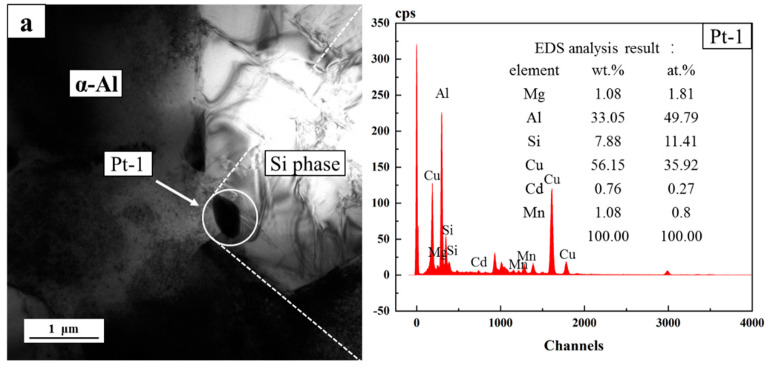
EDS analysis of precipitates near Si phase in Cd-containing alloys after solute treatment: (**a**) two-stage-Cd-containing alloys and (**b**) three-stage-Cd-containing alloys.

**Figure 12 materials-15-05101-f012:**
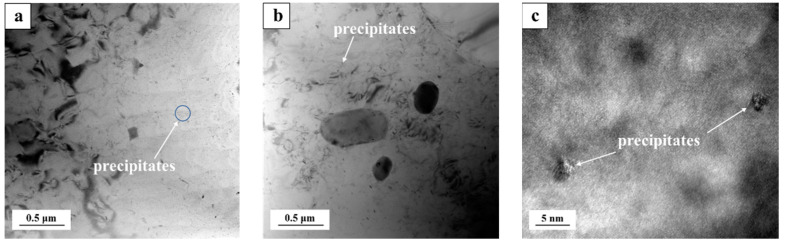
Bright-field TEM diagram of Cd-containing alloys after heat treatment: (**a**) two-stage solution treatment, (**b**) three-stage solution treatment, and (**c**) high-resolution TEM of precipitates.

**Table 1 materials-15-05101-t001:** Composition of different alloys wt.%.

Alloy	Si	Cu	Mg	Mn	Ti	Fe	Sr	Cd	Al
Cd-free	7.52	1.63	0.25	0.25	0.18	0.05	0.06	-	Balance
Cd-containing	7.28	1.57	0.28	0.23	0.19	0.05	0.06	0.29	Balance

**Table 2 materials-15-05101-t002:** Multi-stage solution heat treatment and aging treatment table.

Process of Treatment	Solid Solution Stages						Aging Stage	
	Stage 1		Stage 2		Stage 3			
	T, (°C)	t, (h)	T, (°C)	t, (h)	T, (°C)	t, (h)	T, (°C)	t, (h)
AHT-2	500	4	520	4	-	-	175	-
AHT-3	490	4	500	6	520	8		

**Table 3 materials-15-05101-t003:** Mechanical properties of the Cd-free and Cd-containing alloys after T6 heat treatment.

Alloys	YS (MPa)	UTS (MPa)	El (%)
two-stage-Cd-free	240 ± 3	353 ± 9	11 ± 3
three-stage-Cd-free	279 ± 6	373 ± 7	10 ± 2
two-stage-Cd-containing	308 ± 5	390 ± 7	7 ± 2
three-stage-Cd-containing	295 ± 4	384 ± 4	10 ± 1

## Data Availability

Not applicable.

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
