# Peer review of "Effect of Cd on Mechanical Properties of Al-Si-Cu-Mg Alloys under Different Multi-Stage Solution Heat Treatment"

_materials, 2022, doi:10.3390/ma15155101_

Round 1
Reviewer 1 Report
The manuscript titled "Effect of Cd on properties of Al-Si-Cu-Mg alloys under different multi-stage solution heat treatment" is a good compilation of work. However, there are many similar articles and a few same theme articles related to this paper. The authors need to address the below queries before this articles can be considered for acceptance:
1. The title says "Effect of Cd on properties...". The authors need to mention the specific properties, else it would be vogue.
2. The manuscript needs extensive language corrections. There are some badly framed sentences such as in abstract itself, i.e. "Cu-rich phase full dissolved...".
3. In the line no.18 in the abstract, ""...three-stage heat treatment lead to Cd segregation and precipitation of Cd-rich metal compounds, that reduced the Cd content available for promoting precipitation". The statement gives a strange meaning. Whether Cd has formed an undesirable phase so that the precipitation was not good. The authors need to give clear statements.
4. Please avoid the initials of the authors, while citing in the running script (such as in lines no.59, 68, etc.).
5. Reference 9 is almost similar work done. The authors need to provide a clear distinction between the present paper comparing the previously published papers.
6. A good number of published papers are there on Cd addition to Al alloys, which are not discussed such as ref 7, 9, etc.
7. It would be better to have a phase diagram to show whether Cd can form precipitates are not?
8. In Fig.2, why there is a serious drop in hardness at around 15 h of aging treatment? Please explain in detail.
9. In Fig.5, There is a decent difference in mechanical properties between 2-state and 3-state Cd-free alloys, whereas there is no difference at all for the Cd-containing cases. Why?
10. Try to avoid decimals while projecting tensile properties and better to provide a statistical error bar to represent the reproducibility of the results.
11. Section 4.2 says Cd segregation, whereas Fig.8 doesn't show any Cd segregation at all.
12. Some references are incomplete in Ref section such as Ref. 2, 7, 10, 14, 26, (Overall the reference style is not as per the strict guidelines of the journal)
13. The author names are not appropriate in Ref.19, 17, 40.
14. Conclusions must be changed accordingly.
15. In line no.95, it says "The tensile rate was 0.5 mm/min...). Is this pulling rate or strain rate? Based on units it must be pulling rate, but it is advisable to represent it as strain rate.
I can recommend MAJOR revisions of this paper.
Author Response
Dear Editor:
Thank you for your reply to my article " Effect of Cd on mechanical properties of Al-Si-Cu-Mg alloys under different multi-stage solution heat treatment" and the opinions of external audit experts. We have carefully studied their opinions and made corresponding corrections, hoping to get your approval. The modified part has been highlighted. The amendments are as follows:
Questions for Reviewer #1:
Question [1]: The title says "Effect of Cd on properties...". The authors need to mention the specific properties, else it would be vogue.
Modification result: We replaced Title " Effect of Cd on properties of Al-Si-Cu-Mg alloys under different multi-stage solution heat treatment " with Title " Effect of Cd on mechanical properties of Al-Si-Cu-Mg alloys under different multi-stage solution heat treatment ".
Question [2]: The manuscript needs extensive language corrections. There are some badly framed sentences such as in abstract itself, i.e. "Cu-rich phase full dissolved...".
Modification result: We have polished the language.
Question [3]: In the line no.18 in the abstract, ""...three-stage heat treatment lead to Cd segregation and precipitation of Cd-rich metal compounds, that reduced the Cd content available for promoting precipitation". The statement gives a strange meaning. Whether Cd has formed an undesirable phase so that the precipitation was not good. The authors need to give clear statements.
Thank you for your carefully review, we have noticed this mistake and have modified the relevant sentence as follow.
Modification result: For cd-containing alloys, the three-stage heat treatment led to the precipitation of cd-rich low melting point phases, caused defects and reduced the mechanical properties of the alloy.
Question [4]: Please avoid the initials of the authors, while citing in the running script (such as in lines no.59, 68, etc.).
Modification result: We made changes to the author's name:
- Han et al. → Han et al.
- Aguilera Luna et al. → Aguilera Luna et al.
J.H. Sokolowski et al. → Sokolowski et al.
Runxia, L.I. → Runxia Li et al.
Question [5]: Reference 9 is almost similar work done. The authors need to provide a clear distinction between the present paper comparing the previously published papers.
We had added references 7 and 9 to the difference between this article
Modification result: Hardy et al. [7] and Runxia Li et al. [9] studied the effect of Cd on the aging process of al-Si-Cu-Mg alloy cast aluminum alloy and found that the addition of Cd accelerated the aging process of Al-Si-Cu-Mg alloy. But no one studied the effect of Cd addition on the solid solution process of Al-Si-Cu-Mg alloy. Therefore, it was necessary to study the effect of Cd on Al-Si-Cu-Mg alloys at different solution temperatures and times.
Question [6]: A good number of published papers are there on Cd addition to Al alloys, which are not discussed such as ref 7, 9, etc.
Modification result: We added a discussion about references 7 and 9.
Question [7]: It would be better to have a phase diagram to show whether Cd can form precipitates are not?
Modification result: We added a phase diagram and analyzed it
Question [8]: In Fig.2, why there is a serious drop in hardness at around 15 h of aging treatment? Please explain in detail.
Modification result: There were two ageing peaks in the ageing hardening process of Al-Si-Cu-Mg alloy, the GP region and the metastable phase could effectively strengthen the alloy and led to the ageing peak. In the early stage of alloy aging, fine and abundant GP region was distributed in the supersaturated solid solution (matrix) of the alloy, which played a strengthening role. In the middle stage of aging, the formed metastable phase that maintains a semi-coherent phase with the matrix effectively resisted the movement of dislocations and had a certain strengthening effect [9, 10]. The GP region was known to dissolve significantly before the formation of the metastable phase, and θ phase had been observed to nucleate on dislocations [34]. Therefore, in the stage of transition from the GP region to the metastable phase, the number of GP region was significantly reduced and dissolved, while the metastable precipitates did not grow up and were too small in size to effectively resist the movement of dislocations, resulting in a low age-hardening effect between the two aging peaks of the Al-Si-Cu-Mg alloy.
Question [9]: In Fig.5, There is a decent difference in mechanical properties between 2-state and 3-state Cd-free alloys, whereas there is no difference at all for the Cd-containing cases. Why?
Modification result: Thanks for their careful review. After our inspection, the stretching curve selected for this work was indeed problematic, and this was corrected.
Question [10]: Try to avoid decimals while projecting tensile properties and better to provide a statistical error bar to represent the reproducibility of the results.
Modification result: We changed decimals to whole numbers and added statistical error bars.
Question [11]: Section 4.2 says Cd segregation, whereas Fig.8 doesn't show any Cd segregation at all.
Modification result: Thank you for your careful review, it was indeed our mistake. During the previous study, there was a trend of Cd segregation, but the surface scan results did not show Cd segregation, but a Cd-rich low-melting phase was formed, so we revised the content about Cd segregation in this paper.
Question [12]: Some references are incomplete in Ref section such as Ref. 2, 7, 10, 14, 26, (Overall the reference style is not as per the strict guidelines of the journal)
Modification result: We had made changes to the bibliography format
Question [13]: The author names are not appropriate in Ref.19, 17, 40.
Modification result: We had made changes to the reference author format
Question [14]: Conclusions must be changed accordingly.
Modification result: We have revised our conclusions:
- YS and UTS of Cd-containing alloys are stronger than Cd-free alloys, but El is lower than Cd-free alloys.
- The addition of Cd promotes the precipitation of the θ" phase in the Al-Si-Cu-Mg alloy and improves the mechanical properties of the alloy.
- For Cd-free alloys, long solution time is conducive to the full diffusion and homogenization of copper elements, thereby improving the precipitation strengthening ability of the alloy. For Cd-containing alloys, Cd element is enriched in the Si-Al phase boundary, which reduces the precipitation strengthening ability of Cd, resulting in the alloy's mechanical properties after three-stage heat treatment being lower than that of two-stage heat treatment.
Question [15]: In line no.95, it says "The tensile rate was 0.5 mm/min...). Is this pulling rate or strain rate? Based on units it must be pulling rate, but it is advisable to represent it as strain rate.
Modification result: We convert the stretch rate to the initial strain rate.
I'm very pleased to learn that the manuscript can be published in materials magazine after modification.

Reviewer 2 Report
Authors present a well written and very interesting manuscript, with outstanding results related to the effect of Cd in quaternary Al-Si-Cu-Mg alloys, which can contain different phases and precipitates. Besides, this alloying element can modify the precipitation process and needed this kind of research. That is why I recommend the paper for its publication. Some suggestions in order to maybe improve the manuscript are:
- In my opinion, it is necessary to first introduce the as cast alloy, with micrographies of their phases, for Cd and Cd-free; an only after that present the modifications after the heat treatments which you present in Fig. 1. This could also show the effect of Cd on the microstructure.
- You comment that Al2Cu completely dissolved, and present OM images. Nevertheless, maybe the dissolution is not complete. Al2Cu has a grey scale similar to Al matrix, being difficult to identify using OM. That is why maybe SEM BSE images could be important in order to corroborate tour asseveration about the complete dissolution of Al2Cu. You are presenting EDS and mappings in Fig. 7, but you did not include Mg or Cu to analyze them. Besides, I am not able to distinguish the Cd rich phases that you are mentioning. Contrarily, Cd is uniformly distributed.
- In these alloys other second phases such as Q and Mg2Si could be obtained after solidification (not only after ageing as precipitates, as you mentioned in page 5). Maybe it is important a better analysis of the as cast and solution heat treated alloys, using both OM and SEM at higher magnifications.
- How can you demonstrate that precipitates correspond to the phases that you are indicating in Fig. 3. ? You are presenting results from literature but in my opinion it is important to presence some evidence, e.g. electron diffraction patterns.
-
Author Response
Dear Editor:
Thank you for your reply to my article " Effect of Cd on mechanical properties of Al-Si-Cu-Mg alloys under different multi-stage solution heat treatment" and the opinions of external audit experts. We have carefully studied their opinions and made corresponding corrections, hoping to get your approval. The modified part has been highlighted. The amendments are as follows:
Questions for Reviewer #2:
Question [1]: In my opinion, it is necessary to first introduce the as cast alloy, with micrographics of their phases, for Cd and Cd-free; an only after that present the modifications after the heat treatments which you present in Fig. 1. This could also show the effect of Cd on the microstructure.
Modification result: We had added and analyzed metallographic images of as-cast cd-free and cd-containing alloys
Question [2]: You comment that Al2Cu completely dissolved, and present OM images. Nevertheless, maybe the dissolution is not complete. Al2Cu has a grey scale similar to Al matrix, being difficult to identify using OM. That is why maybe SEM BSE images could be important in order to corroborate tour asseveration about the complete dissolution of Al2Cu. You are presenting EDS and mappings in Fig. 7, but you did not include Mg or Cu to analyze them. Besides, I am not able to distinguish the Cd rich phases that you are mentioning. Contrarily, Cd is uniformly distributed.
Modification result: Thank you for your careful review, it was indeed our mistake. During the previous study, there was a trend of Cd segregation, but the surface scan results did not show Cd segregation, but a Cd-rich low-melting phase was formed, so we revised the content about Cd segregation in this paper.
Question [3]: In these alloys other second phases such as Q and Mg2Si could be obtained after solidification (not only after ageing as precipitates, as you mentioned in page 5). Maybe it is important a better analysis of the as cast and solution heat treated alloys, using both OM and SEM at higher magnifications.
Modification result: Thank you for your careful review and suggestions, because this study mainly discussed the effect of solid solution and aging process on alloys, and phases such as Q phase and Mg2Si were not the main research content, so there is no in-depth discussion.
Question [4]: How can you demonstrate that precipitates correspond to the phases that you are indicating in Fig. 3.? You are presenting results from literature but in my opinion, it is important to presence some evidence, e.g. electron diffraction patterns.
Modification result: We added diffraction spots with cd-containing alloys precipitates and analyzed them:
Figure 12. TEM diagram of Cd-containing alloys after heat treatment: (a) two-stage solution treatment; (b) three-stage solution treatment; (c) precipitate Fast Fourier Transform (FFT) modes.
To further verify the effect of the enrichment of Cd atoms on the precipitation of the strengthening phases, the microstructure of the Cd-containing alloys after the two/three-stage solution heat treatment and ageing were analyzed. Fig. 12(a) shows the bright-field image of the Cd-containing alloys after the two-stage heat treatment. Fig. 12(b) shows the bright-field image of the Cd-containing alloys after the three-stage heat treatment. There was also a certain number of precipitates evenly distributed in the matrix, but the number was slightly smaller. Fig. 12(c) shows the FFT pattern of the precipitates in the matrix of the Cd-containing alloy sample, and the results showed that the massive precipitates were evenly distributed in the matrix and the number was large, which could con-firm that the precipitates were fine θ″ phases precipitates [44]. Comparing the TEM images of the Cd-containing alloys in the two-stage and three-stage heat treatments, it could be found that, after the three-stage solution heat treatment of Cd-containing alloys, the pre-cipitates in the samples were significantly smaller in size and volume fraction, resulting in a lower strengthening effect than the two-stage solution treatment.
I'm very pleased to learn that the manuscript can be published in materials magazine after modification.

Reviewer 3 Report
The manuscript has been written several years ago. Many of the references are very old. The last reference is from 2009 and it says revised reprints. Please answer some of the vital questions.
1 How did you decide how much percentage of Cd is optimum.
2. please provide the XRD and its detailed interpretations for samples with and without Cd
3. How did you decide the optimized temperature for annealing
4. As you mention there are several reports. kindly compare your values with the reported ones to prove that novelty of the article.
5. Why your references are very old?
Author Response
Dear Editor:
Thank you for your reply to my article " Effect of Cd on mechanical properties of Al-Si-Cu-Mg alloys under different multi-stage solution heat treatment" and the opinions of external audit experts. We have carefully studied their opinions and made corresponding corrections, hoping to get your approval. The modified part has been highlighted. The amendments are as follows:
Questions for Reviewer #3:
Question [1]: How did you decide how much percentage of Cd is optimum.
Modification result: According to the aging hardening curve, there is a bimodal aging result, and combined with the research results of Runxia Li et al. [35], it is concluded that the optimal aging process in this experiment is 175℃×12h.
Question [2]: please provide the XRD and its detailed interpretations for samples with and without Cd
Modification result: We provided the XRD and its detailed interpretations for samples with and without Cd
Question [3]: How did you decide the optimized temperature for annealing
Modification result: We previously studied the age-hardening curves at ageing temperatures of 155°C, 175°C, and 200°C, and found that the first peak of hardness at 175°C was higher than the hardness values at the other two temperatures. And by compared the mechanical properties of the alloy after different heat treatment parameters, it was found that the aging process of 175℃×12h was used to obtain the better mechanical properties of the alloy.
Question [4]: As you mention there are several reports. kindly compare your values with the reported ones to prove that novelty of the article.
Modification result: Hardy et al. and Runxia Li et al. studied the effect of Cd on the aging process of cast Al alloy, and found that the addition of Cd could hinder the formation of the GP zone, thereby accelerating the precipitation of the metastable phase, forming a fine and dense metastable precipitation phase on the alloys matrix, which played a role in accelerating the aging process of Al-Si-Cu-Mg alloys. No one had studied the effect of Cd on casting Al-Si-Cu-Mg alloy with different solution times and temperatures. Therefore, it was necessary for us to study the effect of Cd on the Al-Si-Cu-Mg alloy at different solution temperatures and times.
Question [5]: Why your references are very old?
Modification result: Thanks for your careful review, we had brought this issue to our attention and had added some up-to-date references.
I'm very pleased to learn that the manuscript can be published in materials magazine after modification.

Reviewer 4 Report
1. The effective amount of Cd cannot be predicted using only the simple chemical compositions of 'Cd-free' and 'Cd-containing'.
2. As for the heat treatment and aging treatment conditions, only AHT-2 and AH3-3 are considered, so the data are insufficient to derive academic content.
3. In Figure 2, it is shown that the hardness changes for two test condition in the same trend with time. It would be good if it was displayed along with a detailed explanation of what occurred at about 10 hours and 20 hours. The trend needs to be understood.
4. If there is a change in hardness as shown in Figure 2, can the tensile strength value change significantly depending on the heat treatment time? Then it is necessary to show the test results in the intermediate stage of heat treatment.
5. At the tensile test curve in Fig. 5, a common peak is observed for all specimens near 1% engineering strain. It would be nice to have an explanation for this phenomenon.
6. It seems that the contents described in Conclusions 1, 2, and 3 cannot be easily derived from the examination of the main text alone. It would be better to supplement the data by comparing it with the values ​​presented in other existing studies or by presenting more actual measured data.
Author Response
Dear Editor:
Thank you for your reply to my article " Effect of Cd on mechanical properties of Al-Si-Cu-Mg alloys under different multi-stage solution heat treatment" and the opinions of external audit experts. We have carefully studied their opinions and made corresponding corrections, hoping to get your approval. The modified part has been highlighted. The amendments are as follows:
Questions for Reviewer #4:
Question [1]: The effective amount of Cd cannot be predicted using only the simple chemical compositions of 'Cd-free' and 'Cd-containing'.
Modification result: This paper studied the effect of Cd addition on Al-Si-Cu-Mg alloy under different heat treatment parameters, not the effect of optimal Cd content on Al-Si-Cu-Mg alloy. The optimum Cd content could be derived from the phase diagram, which would be discussed in subsequent studies.
Question [2]: As for the heat treatment and aging treatment conditions, only AHT-2 and AH3-3 are considered, so the data are insufficient to derive academic content.
Modification result: In this study, the solution temperature was calculated according to the phase diagram, and the solution time was determined by observing the changes in the microstructure. This study was to explore the effects of the second-level solution heat treatment and the third-level solution heat treatment on the structure and properties of the alloy.
Question [3]: In Figure 2, it is shown that the hardness changes for two test condition in the same trend with time. It would be good if it was displayed along with a detailed explanation of what occurred at about 10 hours and 20 hours. The trend needs to be understood.
Modification result: There were two ageing peaks in the ageing hardening process of Al-Si-Cu-Mg alloy, the GP region and the metastable phase could effectively strengthen the alloy and led to the ageing peak. In the early stage of alloy aging, fine and abundant GP region was distributed in the supersaturated solid solution (matrix) of the alloy, which played a strengthening role. In the middle stage of aging, the formed metastable phase that maintains a semi-coherent phase with the matrix effectively resisted the movement of dislocations and had a certain strengthening effect [9, 10]. The GP region was known to dissolve significantly before the formation of the metastable phase, and θ phase had been observed to nucleate on dislocations [34]. Therefore, in the stage of transition from the GP region to the metastable phase, the number of GP region was significantly reduced and dissolved, while the metastable precipitates did not grow up and were too small in size to effectively resist the movement of dislocations, resulting in a low age-hardening effect between the two aging peaks of the Al-Si-Cu-Mg alloy.
Question [4]: If there is a change in hardness as shown in Figure 2, can the tensile strength value change significantly be depending on the heat treatment time? Then it is necessary to show the test results in the intermediate stage of heat treatment.
Modification result: Since the trend of the hardness curve corresponded to the trend of the tensile properties, this study directly investigated the mechanical properties under the premise of maximum hardness.
Question [5]: At the tensile test curve in Fig. 5, a common peak is observed for all specimens near 1% engineering strain. It would be nice to have an explanation for this phenomenon.
Modification result: As can be seen from Fig. 6, when the engineering strain is 1%, the engineering stress decreases and a relaxation phenomenon occurs because the extensometer is removed here, resulting in stress jitter at the 1% strain position. We manipulated the data and recreated the stretch plots.
Question [6]: It seems that the contents described in Conclusions 1, 2, and 3 cannot be easily derived from the examination of the main text alone. It would be better to supplement the data by comparing it with the values presented in other existing studies or by presenting more actual measured data.
Modification result: We have revised our conclusions:
- YS and UTS of Cd-containing alloys are stronger than Cd-free alloys, but El is lower than Cd-free alloys.
- The addition of Cd promotes the precipitation of the θ" phase in the Al-Si-Cu-Mg alloy and improves the mechanical properties of the alloy.
- For Cd-free alloys, long solution time is conducive to the full diffusion and homogenization of copper elements, thereby improving the precipitation strengthening ability of the alloy. For Cd-containing alloys, Cd element is enriched in the Si-Al phase boundary, which reduces the precipitation strengthening ability of Cd, resulting in the alloy's mechanical properties after three-stage heat treatment being lower than that of two-stage heat treatment.
I'm very pleased to learn that the manuscript can be published in materials magazine after modification.

Round 2
Reviewer 1 Report
The manuscript is really improved, however, still serious fall backs are seen. Especially reference section is still not as per the guidelines of the journal. Further, only 45 references are cited in the manuscript, whereas the reference number crossed 52 (beyond 52 is a mess).
TEM image quality in Fig.12 is poor.
How the authors could conclude that Cd-rich phase is low melting?
In my view, the indexing of XRD pattern is incorrect. Especially, the first peak is supposed to be Al (not Si). It is seriously advised to reindex the XRD patterns and also advised to provide the XRD file number in the manuscript or in the image.
In Fig.7, it is observed that small particles in the dimples in the Cd-containing alloy, which is not discussed seriously.
As per Fig.6 and Table 3, the two-stage treated Cd-free alloy is showing low strength than the three-stage treated Cd-free alloy. Whereas, in the case of Cd-containing alloy two-stage treatment is showing high strength. The authors didn't discuss it in detail. If I am not wrong, the fracture behavior also can be changed because of this. Whereas the authors provided only two conditions of fractography only. Why?
It is advised to follow superscript/subscript details carefully, one such example is in line no.96. (10-4).
Author Response
Dear Editor:
Thank you for your reply to my article " Effect of Cd on mechanical properties of Al-Si-Cu-Mg alloys under different multi-stage solution heat treatment" and the opinions of external audit experts. We have carefully studied their opinions and made corresponding corrections, hoping to get your approval. The modified part has been highlighted. The amendments are as follows:
Questions for Reviewer #1:
Question [1]: The manuscript is really improved; however, still serious fall backs are seen. Especially reference section is still not as per the guidelines of the journal. Further, only 45 references are cited in the manuscript, whereas the reference number crossed 52 (beyond 52 is a mess).
Modification result: We had carefully revised the format of the reference in accordance with the guidelines of the journal.
Question [2]: TEM image quality in Fig.12 is poor.
Modification result: We replaced poor quality images with clearer TEM images:
Figure 12. Bright-field TEM diagram of Cd-containing alloys after heat treatment: (a) two-stage solution treatment; (b) three-stage solution treatment; (c) High-resolution TEM of precipitates.
To further verify the effect of the enrichment of Cd atoms on the precipitation of the strengthening phases, the microstructure of the Cd-containing alloys after the two/three-stage solution heat treatment and ageing were analyzed. Fig. 12(a) shows the bright-field image of the Cd-containing alloys after the two-stage heat treatment. Fig. 12(b) shows the bright-field image of the Cd-containing alloys after the three-stage heat treatment. There was also a certain number of precipitates evenly distributed in the matrix, but the number was slightly smaller. Fig. 12(c) shows the high-resolution TEM of precipitates in the matrix of the Cd-containing alloy sample, and the results showed that the massive precipitates were evenly distributed in the matrix and the number was large, which could confirm that the precipitates were fine θ″ phases precipitates [45]. Comparing the TEM images of the Cd-containing alloys in the two-stage and three-stage heat treatments, it could be found that, after the three-stage solution heat treatment of Cd-containing alloys, the precipitates in the samples were significantly smaller in size and volume fraction, resulting in a lower strengthening effect than the two-stage solution treatment.
Question [3]: How the authors could conclude that Cd-rich phase is low melting?
Modification result: We carried out EDS analysis on the white particles in the fracture, and combined with the research of Runxia et al. [9], we concluded that the Cd-rich phase is a low melting phase. The specific analysis was as follows:
Fig. 7(c) showed that part of the fracture mode was intergranular-fracture, and there were also a large number of dimples, and a white residual phase appeared at the bottom of the dimples. However, Fig. 7(d) showed that a small amount of cleavage platform appears, the tear ridges were obvious, and the dimple became deeper, indicating that the matrix was effectively strengthened. There were more small white particles in Fig. 7(d) than in Fig. 7(c). After EDS analysis, they were mainly Cd-rich phases, indicating that Cd-containing alloys would precipitate more Cd-rich phases with the prolongation of solid solution time. According to the research of Runxia et al. [9], that the Cd-rich phase was a low-melting phase.
Question [4]: In my view, the indexing of XRD pattern is incorrect. Especially, the first peak is supposed to be Al (not Si). It is seriously advised to reindex the XRD patterns and also advised to provide the XRD file number in the manuscript or in the image.
Modification result: We re-indexed the XRD patterns and believed that the first peak was indeed Si, with Al phase XRD file number PDF#65-2869 and Si phase XRD file number PDF#65-1060.
Question [5]: In Fig.7, it is observed that small particles in the dimples in the Cd-containing alloy, which is not discussed seriously.
Modification result: We carried out EDS analysis on the white particles in the fracture. The specific analysis was as follows:
Fig.7 shows the fracture morphologies of Cd-free and Cd-containing alloys after the two-stage solution heat treatment and the 12-hour ageing treatment. Ductile fracture happened in all alloys according to Fig. 7, a large number of small and shallow dimples could be observed in Fig. 7(a), which was reflected in the fracture elongation of the alloy, reaching 11. Fig. 7(b) showed that the local dimples were larger, the dimples are deepened, and the elongation rate was reduced to a certain extent. Fig. 7(c) showed that part of the fracture mode was intergranular-fracture, and there were also a large number of dimples, and a white residual phase appeared at the bottom of the dimples. However, Fig. 7(d) showed that a small amount of cleavage platform appears, the tear ridges were obvious, and the dimple became deeper, indicating that the matrix was effectively strengthened. There were more small white particles in Fig. 7(d) than in Fig. 7(c). After EDS analysis, they were mainly Cd-rich phases, indicating that Cd-containing alloys would precipitate more Cd-rich phases with the prolongation of solid solution time. According to the research of Runxia et al. [9], that the Cd-rich phase was a low-melting phase. As the number of precipitates of the Cd-rich phase increased, it would reduce the grain boundary strength, which was not conducive to the strengthening of the alloy and reduced the mechanical properties, which was consistent with the results of tensile strength.
Question [6]: As per Fig.6 and Table 3, the two-stage treated Cd-free alloy is showing low strength than the three-stage treated Cd-free alloy. Whereas, in the case of Cd-containing alloy two-stage treatment is showing high strength. The authors didn't discuss it in detail. If I am not wrong, the fracture behavior also can be changed because of this. Whereas the authors provided only two conditions of fractography only. Why?
Modification result: We added and analyzed new fracture images. The specific analysis was as follows:
Fig.7 shows the fracture morphologies of Cd-free and Cd-containing alloys after the two-stage solution heat treatment and the 12-hour ageing treatment. Ductile fracture happened in all alloys according to Fig. 7, a large number of small and shallow dimples could be observed in Fig. 7(a), which was reflected in the fracture elongation of the alloy, reaching 11. Fig. 7(b) showed that the local dimples were larger, the dimples are deepened, and the elongation rate was reduced to a certain extent. Fig. 7(c) showed that part of the fracture mode was intergranular-fracture, and there were also a large number of dimples, and a white residual phase appeared at the bottom of the dimples. However, Fig. 7(d) showed that a small amount of cleavage platform appears, the tear ridges were obvious, and the dimple became deeper, indicating that the matrix was effectively strengthened. There were more small white particles in Fig. 7(d) than in Fig. 7(c). After EDS analysis, they were mainly Cd-rich phases, indicating that Cd-containing alloys would precipitate more Cd-rich phases with the prolongation of solid solution time. According to the research of Runxia et al. [9], that the Cd-rich phase was a low-melting phase. As the number of precipitates of the Cd-rich phase increased, it would reduce the grain boundary strength, which was not conducive to the strengthening of the alloy and reduced the mechanical properties, which was consistent with the results of tensile strength.
Question [7]: It is advised to follow superscript/subscript details carefully, one such example is in line no.96. (10-4).
Modification result: We checked and modified the font format in the text.
I'm very pleased to learn that the manuscript can be published in materials magazine after modification.

Reviewer 2 Report
I really appreciate the author´s response. They improved the manuscript fulfilling some requirements which allow to better present they results. Please revese again the manuscript to avoid small mistakes. e.g. "For cd-containing alloys .., of cd-rich low" in abstract. Cd with capital letters. Other case: s 2.38×10-4 (-4 upper case)... reference 9. Runxia et al... not Runxia Lee (page 1 line 43).....
Author Response
Dear Editor:
Thank you for your reply to my article " Effect of Cd on mechanical properties of Al-Si-Cu-Mg alloys under different multi-stage solution heat treatment" and the opinions of external audit experts. We have carefully studied their opinions and made corresponding corrections, hoping to get your approval. The modified part has been highlighted. The amendments are as follows:
Questions for Reviewer #2:
Question[1] :I really appreciate the author´s response. They improved the manuscript fulfilling some requirements which allow to better present they results. Please revese again the manuscript to avoid small mistakes. e.g. "For cd-containing alloys, of cd-rich low" in abstract. Cd with capital letters. Other case: s 2.38×10-4 (-4 upper case) ... reference 9. Runxia et al... not Runxia Lee (page 1 line 43) ....
Modification result: We checked and modified the font format in the text.
I'm very pleased to learn that the manuscript can be published in materials magazine after modification.

Reviewer 3 Report
Question [1]: How did you decide how much percentage of Cd is optimum. from first review is still unanswered.
I am asking about the optimized amount of Cd, how to know how much is optimum and less and more would degrade the properties
Author Response
Dear Editor:
Thank you for your reply to my article " Effect of Cd on mechanical properties of Al-Si-Cu-Mg alloys under different multi-stage solution heat treatment" and the opinions of external audit experts. We have carefully studied their opinions and made corresponding corrections, hoping to get your approval. The modified part has been highlighted. The amendments are as follows:
Questions for Reviewer #3:
Question [1]: How did you decide how much percentage of Cd is optimum. from first review is still unanswered.
I am asking about the optimized amount of Cd, how to know how much is optimum and less and more would degrade the properties
Modification result: Hardy et al. [7] and Runxia et al. [9] studied the effect of Cd on the aging process of Al-Si-Cu-Mg alloy cast aluminum alloy and found that adding Cd to the Al-Si-Cu-Mg alloy could accelerate the aging process of the alloy, and with the increase of Cd addition. The strength of the Al-Si-Cu-Mg alloy showed a trend of first increase and then decrease, and the best effect was achieved when the Cd content was 0.3% [9, 25].
I'm very pleased to learn that the manuscript can be published in materials magazine after modification.

Reviewer 4 Report
It seems that the author has faithfully performed the supplementation of the manuscript. I agree to the publication of the research manuscript.
Author Response
Dear Editor:
Thank you for your reply to my article " Effect of Cd on mechanical properties of Al-Si-Cu-Mg alloys under different multi-stage solution heat treatment" and the opinions of external audit experts. We have carefully studied their opinions and made corresponding corrections, hoping to get your approval. The modified part has been highlighted. The amendments are as follows:
Questions for Reviewer #4:
Question [1]: It seems that the author has faithfully performed the supplementation of the manuscript. I agree to the publication of the research manuscript.
Modification result: Thank you very much.
I'm very pleased to learn that the manuscript can be published in materials magazine after modification.
